# A Novel RPWN Selective Harmonic Elimination Method for Single-Phase Inverter

**Guohua Li [1,2,*], Chunwu Liu [1], Zhenfang Fu [1] and Yufeng Wang [1]**

[1] Faculty of Electrical and Control Engineering, Liaoning Technical University, Huludao 125105, Liaoning, China; lcw19950204@163.com (C.L.); fuzhenfanglntu@163.com (Z.F.); wyf792@163.com (Y.W.)

[2] College of Mechanical Engineering, Liaoning Technical University, Fuxin 123000, China

[*] Correspondence: liguohua@lntu.edu.cn

**Abstract:** In the existing random pule width modulation (RPWM) selective harmonic elimination methods, the formula of switching cycle $T_{N+1}$ is complex, and the duty ratio $D_{N+1}$ of the next switching cycle needs to be calculated in advance. However, in the case of unknown $T_{N+1}$, $D_{N+1}$ is also difficult to calculate accurately, and the two parameters are based on each other. A novel selective harmonic elimination method in RPWM is proposed in this paper. The PWM voltage pulse is placed at the back of the switch cycle, which simplifies the formula of the switch cycle $T_{N+1}$ and eliminates the need to calculate the duty ratio $D_{N+1}$. Two kinds of RPWM selective harmonic elimination ideas are summarized. The general formulas of the switch cycle, the effective random number $k$, and the upper and lower limits of switch frequency corresponding to $k$ are derived. The spectrum shaping of inverter output voltage can be realized without using digital filter in this method. Simple algorithm, small calculation and easy implementation are characteristics of the proposed method. The simulation and experimental results confirm the ability of the proposed method for reducing harmonics at the specific frequency in power spectral density (PSD).

**Keywords:** harmonic; RPWM; selective voltage harmonic elimination; single-phase inverter

---

## 1. Introduction

Random pulse width modulation (RPWM) is an effective method to suppress the electromagnetic interference (EMI) and the electromagnetic vibration and noise of the load [1]. As shown in Figure 1, RPWM can mainly be classified as 1) random switching frequency pulse width modulation (PWM) [2], 2) random pulse position PWM, 3) random switching PWM [3], 4) or hybrid random PWM [4]. According to the randomness of the pulse position, it can be divided into random lead-lag [5], random zero vector [6], random pulse center displacement [7], random pulse position [8], random phase-shifted PWM [9], variable delay random PWM [10], asymmetric carrier random PWM [11], single random pulse position [12], and fractal space vector modulation [13]. However, the traditional RPWM cannot suppress specific harmonics selectively, such as the resonance frequency of motors and other loads. The commonly used selective harmonic elimination pulse width modulation (SHEPWM) [14,15] can eliminate specific harmonics, but it is mainly aimed at low-order harmonics such as $6k \pm 1$ ($k$ is the harmonic order), and has little effect on eliminating the resonance frequency of the loads.

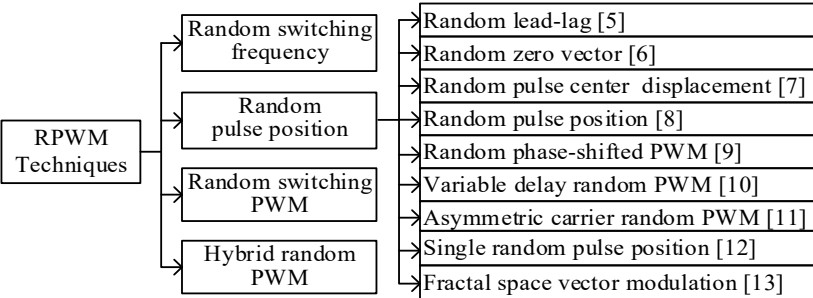

**Figure 1.** Random pulse width modulation (PWM) techniques.

In order to shape the noise spectrum of the inverter output voltage, the method of random modulation for selectively reducing the noise power at one or more frequencies was proposed in [16], but this method led to an increase in the peak of power spectral density. The literature [17] can also reduce the noise power at a specific frequency, but with this method, the switching frequency of the inverter must be less than the resonance frequency. When the resonance frequency is low, the switching frequency of the inverter will be over low. The low-pass filters and band-pass filters are used in [18–20], respectively, to reduce the harmonic power in the specific frequency range. However, the digital filter brings large computation costs in [18–20], and the harmonics in the specific frequency range are avoided, increasing rather than being completely eliminated in the random spread-spectrum. In other words, the harmonic content in the specific frequency range is just not increased compared with the original.

In RPWM selective harmonic elimination method, the specific harmonics can be eliminated by canceling each other with the preceding and succeeding terms in the Fourier series of the output voltage for the inverter. Theoretically, the specific harmonics can be completely eliminated by this method, which is mainly aimed at high-order harmonics such as 7 kHz, 9 kHz, etc. In [21], only harmonics whose frequencies are larger than 20 kHz can be eliminated. However, there is little practical value for the reason that the resonant frequency of the load is mostly lower than 20 kHz. The problem in [21] was solved by the method in [22], and the harmonics whose frequencies are lower than 10 kHz can be eliminated. However, the calculation of $T_{n+1}$ in [22] is complicated, and $D_{n+1}$ needs to be calculated in advance. Because $D_{n+1}$ is calculated according to the midpoint of $T_{n+1}$, $T_{n+1}$ and $D_{n+1}$ are based on each other. In addition, the general formulas of the random number $k$ and its corresponding switching frequency extreme value are not given in [22].

A novel RPWM selective harmonic elimination method for single-phase voltage source inverters (VSI) is proposed. In this method, the PWM pulse is placed at the back of the switching cycle. The calculation of switching cycle $T_{n+1}$ is simplified and the contradiction between $T_{n+1}$ and $D_{n+1}$ is solved. The general formulas of switching cycle and the random number $k$ and its corresponding switching frequency extreme value are also given. The noise of the specific frequency and its multiples can be selectively reduced in this method while realizing the function of traditional RPWM.

## 2. Strategy of Selective Harmonic Elimination in RPWM

The calculation of the switching cycle $T_{n+1}$ is very significant in the RPWM selective harmonic elimination method. The formula for $T_{n+1}$ and its corresponding pulse position in [22] are shown in Equation (1) and Figure 2a, where $D_n$ and $D_{n+1}$ are the duty ratios of the $n$th and $(n+1)$th switching cycles, $k$ is the random number, $f_0$ is the frequency to be eliminated, and $T_n$ is the $n$th switching cycle.

$$T_{n+1} = \frac{2k - f_0 T_n(1 + D_n)}{f_0(1 + D_{n+1})} \tag{1}$$

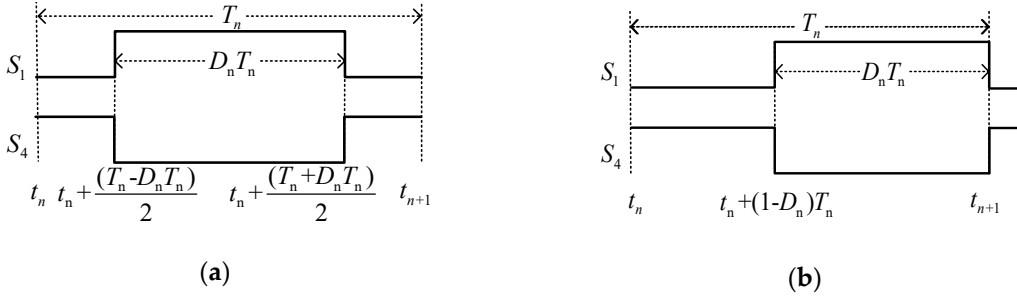

**(a)**

**(b)**

**Figure 2.** The PWM pulse is located at (**a**) the center of switching cycle; (**b**) the back of switching cycle.

The $D_{n+1}$ needs to be calculated in advance to calculate $T_{n+1}$ in Equation (1). In the strategy of SPWM, the duty ratio $D$ is calculated as $(1+M\sin(\omega t))/2$, where $M$ is the modulation ratio. The value of $\omega t$ is calculated according to the time of the midpoint in each switching cycle. Namely, the time of the midpoint in $T_{n+1}$ is used to calculate $D_{n+1}$. Therefore, when $T_{n+1}$ is unknown, the duty ratio $D_{n+1}$ cannot be calculated. Moreover, Equation (1) is complex and there are many parameters in it.

As shown in Figure 2b, the PWM pulse is located at the back of the switching cycle in this paper. This sequence pulse can be regarded as the sum of the output voltage $u_{AB}$ and the DC-link voltage $V_{dc}$ in single-phase VSI, as shown in Figure 3. If there are not the specific harmonics in the sequence pulse spectrum, there are not those harmonics in the output voltage $u_{AB}$ spectrum of single-phase VSI, either.

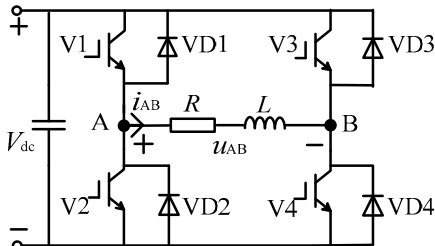

**Figure 3.** Topology of single-phase voltage source inverter.

The equation of the *n*th cycle of the sequence pulse is shown in Equation (2), where $A$ is high value of output voltage and the equation of the sequence pulse is shown in Equation (3), where $t_n$ and $t_{n+1}$ are the starting and ending time of the *n*th switching cycle. The Fourier transform of Equation (3) is given at Equation (4). The real and imaginary parts in Equation (4) are special cases of Equation (5) [15,16]. If $c(f_0)$ at the frequency to be eliminated is zero for any $\varphi$, the result in Equation (4) will also be zero. That is, selective harmonic elimination is realized in the spectrum of the sequence pulse.

$$g_n(t) = \begin{cases} A & t_n+(1-D_n)T_n \leq t < t_{n+1} \\ 0 & otherwise \end{cases} \tag{2}$$

$$g(t) = \lim_{N\to\infty} \sum_{n=1}^{N} g_n(t) \tag{3}$$

$$G(f) = \int_{-\infty}^{\infty} g(t)e^{-j\omega t}dt = \int_{-\infty}^{\infty} g(t)\cos(\omega t)dt - j\int_{-\infty}^{\infty} g(t)\sin(\omega t)dt \tag{4}$$

$$c(f_0) = \int_{-\infty}^{\infty} g(t)\sin(2\pi f_0 t + \varphi)dt \tag{5}$$

## 3. Strategy for Novel RPWM Selective Harmonic Elimination

### 3.1. Calculation for Switching Cycle

Insert Equations (2) and (3) in Equation (5) for calculating Equation (6).

$$c(f_0) = \lim_{N\to\infty} \sum_{m=1}^{N} \left( \int_{t_m+(1-D_m)T_m}^{t_{m+1}} A\sin(2\pi f_0 t + \varphi)dt \right) = \frac{A}{2\pi f_0} \sum_{m=1}^{\infty} \left( \cos\{2\pi f_0\left[t_m + (1-D_m)T_m\right] + \varphi\} - \cos(2\pi f_0 t_{m+1} + \varphi) \right) \quad (6)$$

Two RPWM selective harmonic elimination ideas can be summarized on the basis of Equation (6). The first idea is that the first summation of the *n*th term is removed with the second summation of the (*n*+*e*)th term. The first summation of the (*n*+1)th term is removed with the second summation of the (*n*+*e*+1)th term, etc. The second idea is that the second summation of the *n*th term is removed with the first summation of the (*n*+*e*)th term. The second summation of the (*n*+1)th term is removed with the (*n*+*e*+1)th term of the first summation, etc. The following Equations (7)–(11) are given when *e* is equal to 1 and 2 according to the first idea, where *e* is a positive number.

$$\cos\{2\pi f_0 [t_n + (1-D_n)T_n] + \varphi\} - \cos(2\pi f_0 t_{n+e+1} + \varphi) = 0 \quad (7)$$

$$2\pi f_0 t_{n+e+1} + \varphi = 2\pi f_0[t_n + (1-D_n)T_n] + \varphi + 2k\pi \quad (8)$$

$$T_n = \frac{1}{1-D_n} \left( \sum_{m=n}^{n+e} T_m - \frac{k}{f_0} \right) \quad (9)$$

$$T_{n+1} = \frac{k}{f_0} - D_n T_n \ (e=1) \quad (10)$$

$$T_{n+2} = \frac{k}{f_0} - D_n T_n - T_{n+1} \ (e=2) \quad (11)$$

The comparison between Equation (10) and Equation (1) shows that the calculation of $T_{n+1}$ with the method in this paper is simpler than the method in [22]. In addition, there is no need to calculate $D_{n+1}$. Thereby, the contradiction between $T_{n+1}$ and $D_{n+1}$ is solved and it is conducive to practical application.

### 3.2. Random Number k and Its Corresponding Extreme Value of Switching Frequency

The following Equations ((12) and (13)) for $k_{max}$ and $k_{min}$ are given by using Equation (10) if those conditions ($f_0$, $D_{max}$, $D_{min}$, $f_{ma}$, and $f_{min}$) are given, where $k_{max}$ and $k_{min}$ are the maximum and minimum values of random number *k*. $D_{max}$ and $D_{min}$ are the maximum and minimum values of the duty ratio. $F_{max}$ and $f_{min}$ are the maximum and minimum values of the instantaneous switching frequency of the inverter, which are usually preset.

$$k_{\max} \le \frac{f_0(1 + D_{\max})}{f_{\min}} \quad (12)$$

$$k_{\min} \ge \frac{f_0(1 + D_{\min})}{f_{\max}} \quad (13)$$

Generally, there are many numbers for *k* that satisfy Equations (12) and (13), and the general formulas for $f_{kmax}$ and $f_{kmin}$ corresponding to each *k* are shown in Equations (14) and (15), where $f_{kmax}$ and $f_{kmin}$ are the maximum and minimum values of the switching frequencies corresponding to *k*. It can be seen from Equations (14) and (15) that the switching frequency of the inverter decreases with increasing *k*, or increases with decreasing *k*.

$$f_{k\max} = \frac{1}{\frac{k}{f_0} - \frac{D_{\max}}{f_{\min}}} \quad (14)$$

$$f_{k\min} = \cfrac{1}{\cfrac{k}{f_0} - \cfrac{D_{\min}}{f_{\max}}} \tag{15}$$

It should be noted that the calculation result of Equation (14) may be negative if the $k$ is small. It shows that the denominator can be zero without taking the extreme value of $f$, and the $D$ and the maximum value of the frequency is $+\infty$.

### 3.3. Calculation Process for Selective Harmonic Cancellation

As shown in Figure 4, the frequency to be eliminated ($f_0$) and the maximum value ($f_{max}$) and minimum value ($f_{min}$) of the instantaneous switching frequency are given in advance. The $D_{max}$ and $D_{min}$ are calculated according to the modulation ratio $M$. The range of the random number $k$ can be acquired with the mentioned conditions inserted in Equations (12) and (13). $T_{n+1}$ can be calculated with $T_n$ and $D_n$ inserted in Equation (10) when $k$ is selected. Based on the above conditions, the PWM drive signals are generated by assigning a value to the comparison register in DSP TMS320F2812 to eliminate the harmonics at the specific frequency $f_0$ in RPWM.

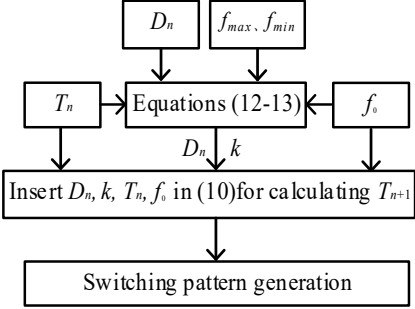

**Figure 4.** Flow chart of switching cycle calculation.

## 4. Simulation and Experiment

### 4.1. Parameters of System

The experimental system is shown in Figure 5. The parameters of the simulation and experimental system are shown in Table 1. The power electronic component in the inverter is IGBT. The driving circuit adopts IGBT-integrated driving module DA962D and the system main control chip adopts 32-bit DSP TMS320F2812. The dead time of inverter is 4.27 μs. In the experiment, the oscilloscope is DS1052E and the power quality analyzer is HIOKI PW3198.

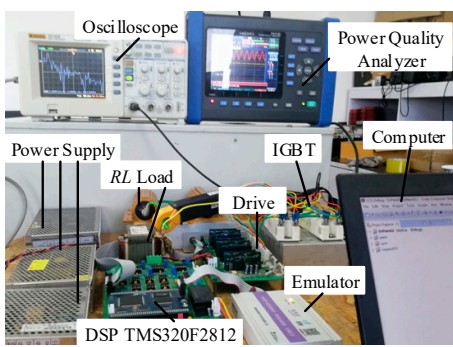

**Figure 5.** Experimental system.

**Table 1.** System parameters.

| Parameter | Value | Parameter | Value |
|-----------|-------|-----------|-------|
| $f_0$(kHZ) | 7 and 9 | $R(\Omega)$ | 5 |
| $F_{max}$(kHZ) | 8 | $L$(mH) | 5 |
| $f_{min}$(kHZ) | 1.5 | DC-link voltage(V) | 24 |

## 4.2. Results and Analysis

The simulation waveforms of output voltage power spectral density (PSD) for single-phase VSI are shown in Figure 6. The traditional SPWM of fixed switching frequency (3 kHz) is adopted in Figure 6a. As seen from Figure 6a, the harmonics are mainly concentrated near 3 kHz and its multiples. Compared with the fixed switching frequency SPWM, it can be seen from Figure 6b with traditional RPWM that there is no outstanding peak in PSD. Figure 6c–e adopt the proposed method in this paper. The frequency to be eliminated ($f_0$) is 7 kHz. The modulation ratio $M$ is 0.9, 0.7, and 0.5, respectively. Harmonics can be distributed randomly and can be reduced greatly at $f_0$ and its multiples.

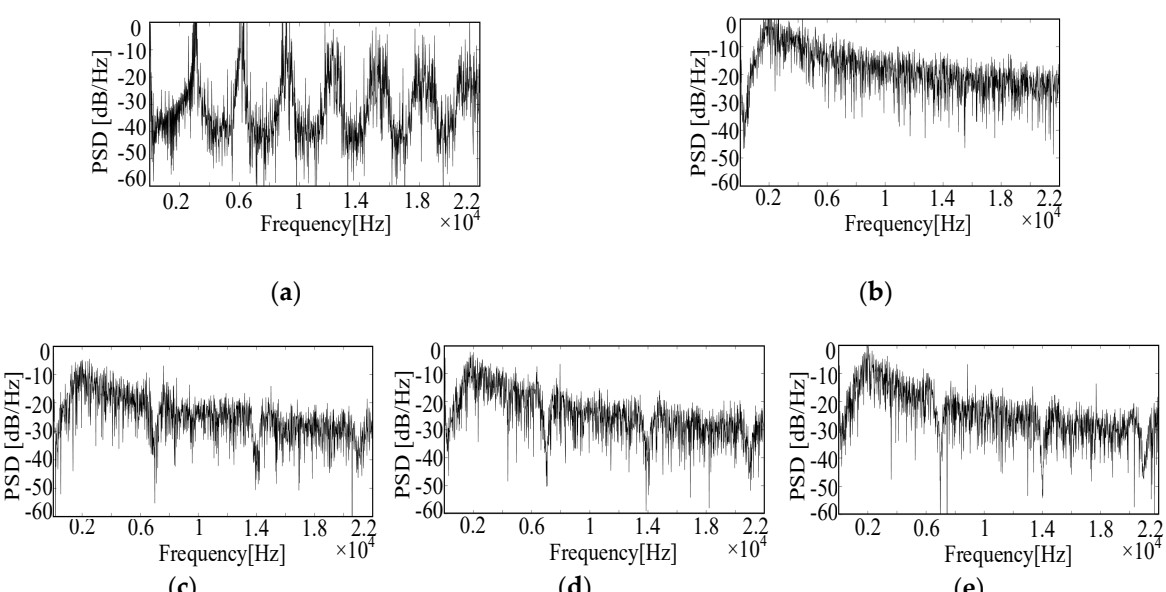

**Figure 6.** Simulated waveforms of voltage power spectral density (PSD). (**a**) Fixed switching frequency; (**b**) Traditional RPWM; and proposed method when $f_0$ = 7 kHz (**c**) $M$ = 0.9; (**d**) $M$ = 0.7; (**e**) $M$ = 0.5.

Figure 7 is the simulation waveforms for the output voltage and current of the single-phase VSI when $M$ is 0.9 and $f_0$ is 7 kHz. The enlarged waveforms of Figure 7 are shown in Figure 8.

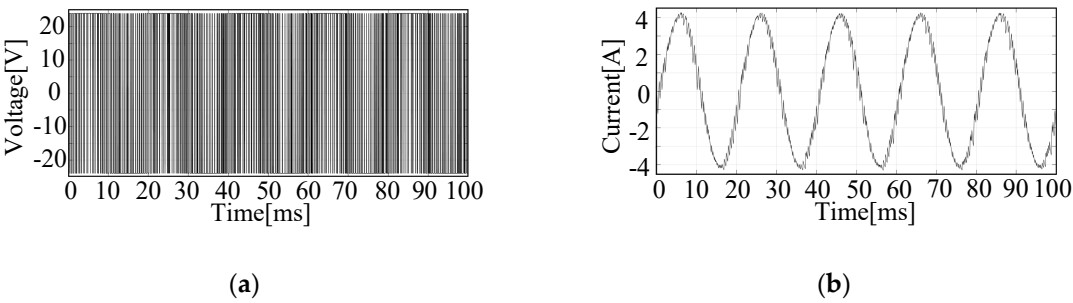

**Figure 7.** Simulated waveforms of single-phase inverter (**a**) voltage; (**b**) current.

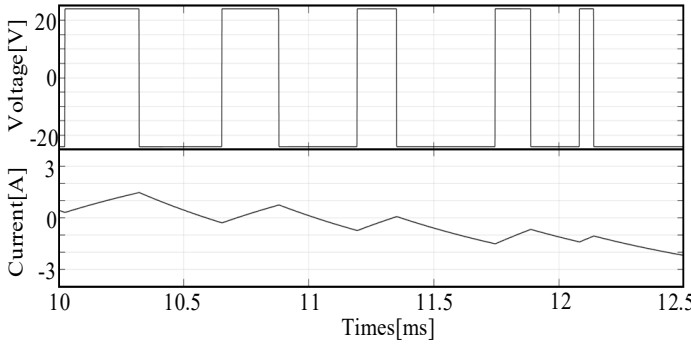

**Figure 8.** Enlarged waveform of Figure 7.

The experimental waveforms of output voltage and current PSD for single-phase VSI are shown in Figure 9. It can be seen that the harmonics near $f_0$ and its multiples are significantly reduced when $f_0$ is taken at 7 kHz and 9 kHz, respectively. The experimental results are basically consistent with the simulation results. As seen from the waveforms in Figures 6 and 9, there are obvious gaps in the range of several hundred hertz around $f_0$ and its multiples. The reason is that the frequencies close to $f_0$ will also be reduced when the harmonic at $f_0$ is completely eliminated. Moreover, the closer it is to $f_0$, the more it is reduced. The influence of system error can be overcome by this characteristic. Thus, the correctness of proposed method is proved by simulation and experimental results.

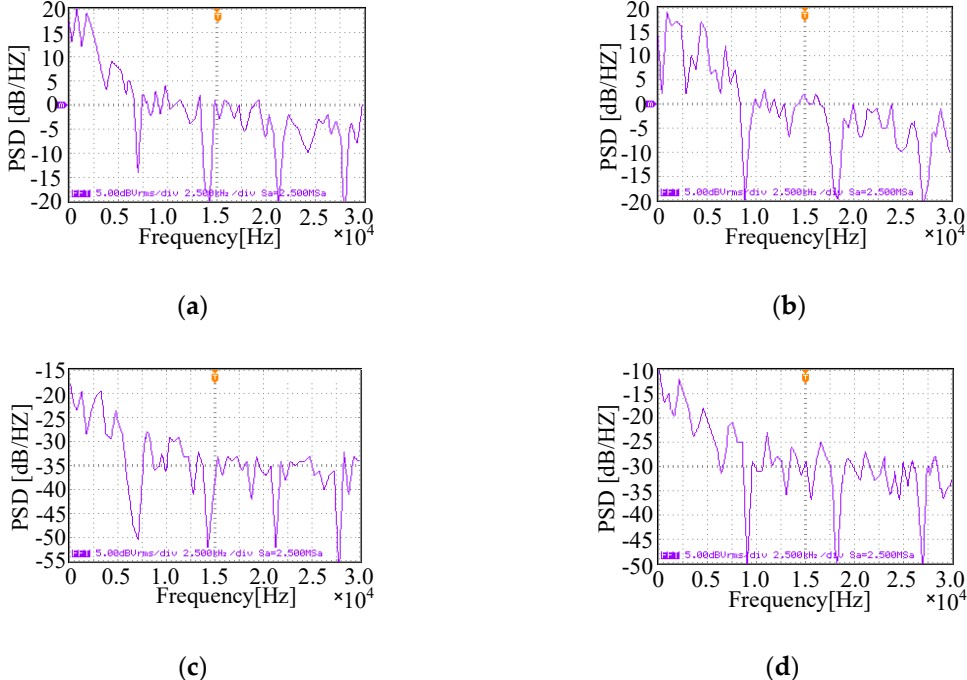

**Figure 9.** Experimental waveforms of PSD when *M* is 0.9. (**a**) The PSD of voltage when $f_0$ is 7 kHz; (**b**) the PSD of voltage when $f_0$ is 9 kHz; (**c**) the PSD of current when $f_0$ is 7 kHz; (**d**) the PSD of current when $f_0$ is 9 kHz.

The distributions of switching frequencies for two set numbers (*k* and $k_1$) when $f_0$ is 7 kHz and *M* is 0.9 are shown in Figure 10. All set random numbers *k* = 1, 2, 3, 4, 5, 6, 7, 8, 9 and smaller set numbers $k_1$ = 1, 2, 3, 4 for single-phase inverter are used for proposed method when switching frequencies are selected to eliminate harmonic at 7 kHz. In Figure 10, the switching frequencies have been selected randomly from 1.5 kHz to 8 kHz and the average switching frequency is equal to 2894 Hz (solid line). The distribution of switching frequencies gradually decreases and the randomness is good. By using $k_1$,

the average switching frequency is increased to 3723 Hz (dot line). It can be known from Equations (14) and (15) that the random number $k$ is inversely proportional to $f_{kmax}$ and $f_{kmin}$. The average switching frequency can be increased by using smaller $k$ when the average switching frequency is low to prevent the increase of current ripples. The average switching frequency can be reduced by using larger $k$ when the average switching frequency is high. In addition, the larger $f_0$ is, the more $k$ can be selected.

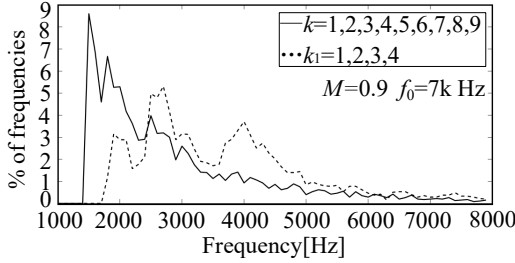

**Figure 10.** Distribution of switching frequencies by using the set numbers of $k$ and $k_1$.

Figure 11 is the experimental waveforms for the output voltage and current of the single-phase VSI when $M$ is 0.9 and $f_0$ is 7 kHz. As can be seen from Figure 11, the voltage pulse width varies randomly in each switching cycle affected by $k$ and $D$. The current waveform of the inverter is sinusoidal.

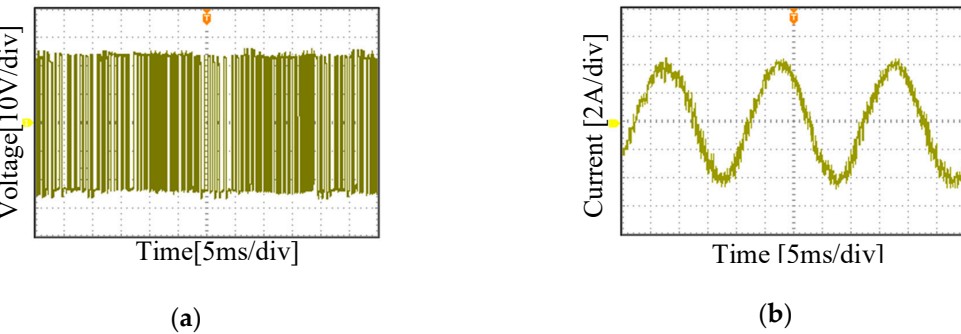

(**a**)          (**b**)

**Figure 11.** Experimental waveforms of single-phase inverter (**a**) voltage; (**b**) current.

The enlarged waveforms of Figure 11 are shown in Figure 12. With other cases unchanged, the larger the switching cycle is, the larger the current ripples are.

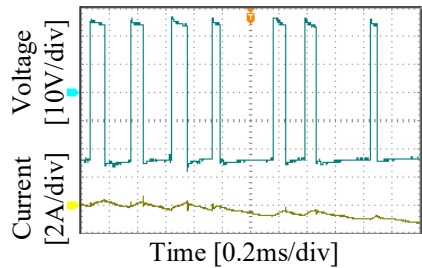

**Figure 12.** Enlarged waveform of Figure 11.

## 5. Conclusions

In this paper, a novel RPWM selective harmonic elimination method for single-phase VSI is proposed. A new pulse position which is placed at the back of the switching cycle is provided for RPWM selective harmonic elimination method. By using that, it can remove harmonics with certain frequency from output voltage and current. In fact, this has been done using switching cycle determination by switching cycles in the previous cycle and duty ratio. Compared with the fixed

switching frequency SPWM and the traditional RPWM, the harmonics can be distributed uniformly in certain frequency range, and some unwanted harmonics can be eliminated successfully with the proposed method. A new pulse position for the RPWM selective harmonic elimination method is introduced in this paper, which is beneficial to improve the randomness of RPWM. When calculating $T_{n+1}$, $D_{n+1}$ does not need to be calculated first. The random arrangement of the pulse position in switching cycle is worth studying in next steps.

**Author Contributions:** Conceptualization, G.L. and C.L.; methodology, Y.W.; software, Z.F.; validation, G.L. and C.L.; formal analysis, Y.W.; writing—original draft preparation, G.L.; writing—review and editing, C.L. All authors have read and agreed to the published version of the manuscript.

**Funding:** This research was funded by National Natural Science Foundation of China, grant number 51307076 and Natural Science Foundation of Liaoning Province, China, grant number 20180550268.

**Acknowledgments:** The authors thank Liaoning Technical University for providing the support to enable this research to be carried out.

**Conflicts of Interest:** The authors declare no conflict of interest.

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
