# Peer review of "A Novel RPWN Selective Harmonic Elimination Method for Single-Phase Inverter"

_electronics, doi:10.3390/electronics9030489_

Round 1
Reviewer 1 Report
In the proposed method, the pulses are placed at the back of the switching cycle, what's the difference with the approach to allocating the pulses at the beginning of the switching period in [22]? Are they equivalent methods? Can authors explain the difference with the proposed method below:Peyghambari, Amir, Ali Dastfan, and Alireza Ahmadyfard. "Strategy for switching period selection in random pulse width modulation to shape the noise spectrum." IET Power Electronics 8, no. 4 (2015): 517-523.
Basically, the concept, derivation process and the conclusion are almost the same.
Can the proposed method be applied to the three-phase inverter? What alterations should be made accordingly? The expression is not complete at line 31, Page 1 Variable k has not been defined at line 35, page 1
Author Response
Dear Reviewer,
Thank you for your comments concerning our manuscript entitled “A Novel RPWM Selective Harmonic Elimination Method for Single-phase Inverter” (ID: electronics-695360). Those comments are all valuable and very helpful for revising and improving our paper, as well as the important guiding significance to our researches. We have studied comments carefully and have made correction which we hope meet with approval. Revised portions and English editing are marked in red in the paper. The revised version has been uploaded as an attachment.
Once again, thank you very much for your comments and suggestions.
Best regards!

Reviewer 2 Report
Authors describe a selective method to eliminate selective harmonic for single-phase inverter. The bibliography is comprehensive enough to highlight the originality of the method proposed in the article. But some errors appear in the manuscript :
please, Can you change the writing of the physical units : kHz ? Table 1 : put the unit into brackets and in the table heads Table 1 : confusion between , and . (dot) for parameters values Legends must be near figuresI understand the method but equations development must be improved :
A is not defined, e is not defined, line 100 e=2 and in equation (9) e=1: why ? confusion between Φ and (small) Φ, author must more explain calculus development in equations (6)-(10) at the first reading k parameter seems to be random number but in equation (7) for me it's modulo. Can you explain this choice ?The experimental system is clearly described ans experimental results give good agreement with the theory.
Author Response

(The authors gave the same response as above.)

Reviewer 3 Report
- Please improve the quality of fig. 6 and 7.
- Please check English. There are multiple typos.
- Please emphasize more clearly the contribution of the paper.
Author Response

(The authors gave the same response as above.)

Round 2
Reviewer 1 Report
The authors have not answer my raised questions properly. What are the difference between the approach presented in the paper and with the proposed method below:
Peyghambari, Amir, Ali Dastfan, and Alireza Ahmadyfard. "Strategy for switching period selection in random pulse width modulation to shape the noise spectrum." IET Power Electronics 8, no. 4 (2015): 517-523.
As what I mentioned, the concept, derivation process and the conclusion are very similar. Authors must explain the difference in details. Also, the unique concept and contribution of the proposed method must be clarified in the manuscript.
Author Response
Dear Reviewer,
Thank you for your comments concerning our manuscript entitled “A Novel RPWM Selective Harmonic Elimination Method for Single-phase Inverter” (ID: electronics-695360). Those comments are all valuable and very helpful for revising and improving our paper, as well as the important guiding significance to our researches. We have studied comments carefully and have made correction which we hope meet with approval.
Once again, thank you very much for your comments and suggestions.
Best regards!

Round 3
Reviewer 1 Report
Thanks for the answers. Authors are advised to have a proofreading throughout the manuscprit before the final submission.